# What Works in Chest X-Ray Classification? A Case Study of Design Choices

Evan Vogelbaum [* 1]   Logan Engstrom [* 1]   Aleksander Madry [1]

## Abstract

Public competitions and datasets have yielded increasingly accurate chest x-ray prediction models. The best such models now match even human radiologists on benchmarks. These models go beyond "standard" image classification techniques, and instead employ design choices specialized for the chest x-ray domain. However, as a result, each model ends up using a different, non-standardized training setup, making it unclear how individual design choices—be it the choice of model architecture, data augmentation type, or loss function—actually affect performance. So, which design choices should we use in practice? Examining a wide range of model design choices on three canonical chest x-ray benchmarks, we find that by simply leveraging a (properly tuned) model composed of up *standard* image classification design choices, one can often match the performance of even the best *domain-specific* models. Moreover, starting from a "barebones," generic ResNet-50 with cross-entropy loss and no data augmentation, we discover that *none* of the proposed design choices—including broadly used choices like the DenseNet-121 architecture or basic data augmentation—consistently improve performance over that generic learning setup.

## 1. Introduction

Public competitions and datasets have led to increasingly better chest x-ray prediction models (Yuan et al., 2020; Pham et al., 2021; Rajpurkar et al., 2017). This leap in performance has been largely fueled by going beyond "standard" choices in computer vision, and instead developing specialized learning methods designed for clinical domains.

This approach has been fruitful: methods using such

"domain-specific" design choices—ranging from pooling layers designed to help localize medical abnormalities during training (Ye et al., 2020) to losses that handle class imbalances (Rajpurkar et al., 2017)—are now the state-of-the-art on chest x-ray classification benchmarks. In fact, the best machine learning models can even *match* radiologist performances on chest x-ray classification (Pham et al., 2021; Yuan et al., 2020; Ye et al., 2020).

However, it is unclear exactly which of these *design choices* actually drive these levels of performance. Indeed, the design choices used by the highest performing models vary widely, making the effect of each *individual* design choice hard to discern. How, then, should we design chest x-ray models in practice?

**Contributions.** We examine models over a wide range of common chest x-ray prediction design choices, spanning data augmentation, training loss, pooling mechanism, and architecture. Specifically, we consider design choices falling in two different categories: "domain-specific" (methods designed for clinical tasks—e.g., a loss that exploits medically known relationships between labels (Pham et al., 2021)) and "standard" (methods used throughout image classification, e.g., cross-entropy loss).

Evaluating on three canonical benchmarks (CHEXPERT (Irvin et al., 2019), MIMIC (Johnson et al., 2019), and CHEST X-RAY 14 (Wang et al., 2017)), we find that domain-specific methods *do not* generally drive performance (compared to standard design choices). In particular, on two of the three benchmarks—CHEXPERT and MIMIC—models using domain-specific design choices *never* improve over the best models employing only standard computer vision methods. On the remaining benchmark—CHEST X-RAY 14—the improvement is *uniformly* less than 2%.

Digging deeper, we try to understand which design choices actually *do* drive chest x-ray model performance. We find that starting from the "barebones" generic ResNet-50 with cross-entropy loss and *no* data augmentation, we *cannot* consistently improve performance by swapping in *any* other singular design choice. We add data augmentation, swap in a DenseNet-121, and even switch to a domain-specific loss or pooling method: *none* of these individual choices commonly thought to improve performance *actually* yield

---

*Equal contribution  [1]Department of Computer Science, MIT, MA, USA. Correspondence to: Evan Vogelbaum <evanv@mit.edu>.

*Workshop on Interpretable ML in Healthcare at International Conference on Machine Learning (ICML)*, Honolulu, Hawaii, USA. 2023.

improvements across all three benchmarks.

Our findings underscore the need to contextualize improvements with uncertainty measures and rigorously compare with strong baselines when presenting new methods. The results also suggest one of two possible scenarios. Either we have not yet developed any design choices that broadly drive performance in the chest x-ray domain (and thus further algorithmic work is needed). Or, we have already discovered such design choices, but existing chest x-ray benchmarks are not *sufficiently representative* of the spectrum of tasks which occur in the chest x-ray domain—for example, the label quality may be too low for improvements to emerge (in which case further dataset construction work is required).

## 2. Design Choices and Methodology

We focus on three canonical chest x-ray classification datasets (CHEXPERT (Irvin et al., 2019), MIMIC (Johnson et al., 2019), and CHEST X-RAY 14 (Wang et al., 2017)) and the best performing methods on these datasets. Our study considers both design choices made specifically for the chest x-ray domain (i.e. *domain-specific* choices) and design choices that originated in general image classification but have worked well on chest x-ray tasks too (i.e. *standard* choices). As a performance baseline, we use a *generic* image classification model whose design choices we describe below. (Note that within this framework, *generic* design choices are a subset of *standard* design choices.)

To ensure we examine the effect of every design choice, we construct our *methods* combinatorially. Each method consists of a choice for each of the following *axes* of a machine learning model, where † indicates a part of the generic model and ⋆ indicates chest x-ray domain-specific choices:

1. **Loss Function**: We consider five different loss functions. (†**1**) BCE: Standard Binary Cross Entropy loss. (**2**) Focal (Lin et al., 2017): A loss function designed to handle class imbalance, a common occurrence in chest x-ray benchmarks. (⋆**3**) CheXNet (Rajpurkar et al., 2017): BCE loss with weights designed to balance the binary labels for each class. (⋆**4**) DAM (Yuan et al., 2020): A novel loss function designed to directly maximize AUROC, the standard metric for chest x-ray classification. (⋆**5**) Hierarchical[1] (Pham et al., 2021): A loss function designed to leverage a medical hierarchy over labels.

2. **Backbone**: We consider five different backbones for

---

[1]We follow Pham et al. (2021) for a hierarchy for the CHEXPERT and MIMIC datasets and construct our own medically valid hierarchy for the CHEST X-RAY 14 dataset. All three are included in Appendix A.1.

our methods. (**1**) ResNet-18, (†**2**) ResNet-50, (**3**) DenseNet-121, (**4**) VGG16, and (**5**) VGG19 with Batch Norm. All of these backbones have been used in chest x-ray classification (Moses, 2021).

3. **Pooling Function**: We consider two different pooling functions for our methods. (†**1**) Standard: The standard pooling function for the backbone of the method as implemented in `torchvision` (Paszke et al., 2019). (⋆**2**) PCAM (Ye et al., 2020): A probabilistic form of class activation maps designed to produce higher performance and interpretability for chest x-ray classification models.

4. **Data Augmentation**: We consider three different data augmentation schemes for our methods. (†**1**) No Aug: No data augmentation is used. (**2**) CIFAR: We randomly translate and cutout a portion of the image. This augmentation is representative of a standard augmentation scheme for the CIFAR-10 dataset (Krizhevsky, 2009). (**3**) ImageNet: We use color jitter, random crop and mixup (Zhang et al., 2017) as a representative of a standard augmentation scheme for the ImageNet dataset (Deng et al., 2009).

We select hyperparameters for each method through grid search. We evaluate methods using average AUROC over classes which is the standard performance metric for chest x-ray classification and report test set scores in our results. We evaluate uncertainty in all values we report by bootstrapping over the test set 1000 times and derive confidence intervals by taking the 2.5% and 97.5% quantiles of the bootstrap distribution. Additional training details can be found in Appendix A.2.

## 3. Evaluation of Design Choices

We begin by examining the performance of domain-specific design choices as compared to standard image-classification choices. We then evaluate the effect of each design choice (both standard and domain-specific) by comparing to the design choices of a generic image classification model. Finally, we examine the effect of each design choice on the core metrics for decision-making: TPR and FPR.

### 3.1. Domain-Specific vs. Standard Design Choices

Many of the design choices made by state-of-the-art methods in chest x-ray classification are domain-specific. However, it is not clear whether domain-specific choices are necessary to achieve strong performance. To evaluate this, we compare the performance of the best methods constructed using domain-specific design choices with the performance of the best methods constructed using only standard design choices. For each domain-specific design choice, we find

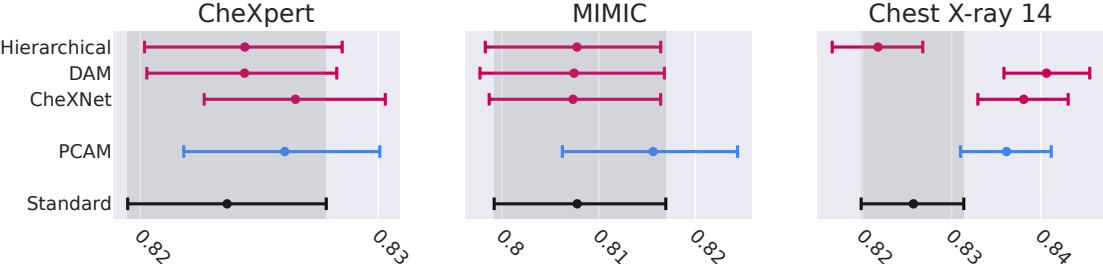

*Figure 1.* Comparing domain-specific design choices with standard ones. For each domain-specific design choice we compare the best method we can construct using that design choice with the best method we can construct using only standard choices. Error bars for test AUROC are computed by bootstrapping the test set 1000 times and recomputing average AUROC for each one. We find that on CHEXPERT and MIMIC the standard method is just as strong as the best methods constructed using domain-specific design choices. On CHEST X-RAY 14, the standard method is not quite as strong, however it is still comparable to Hierarchical Loss, and it is within 1.5% AUROC of other methods.

the choices for other axes of the method which maximize the validation AUROC among methods which use that design choice. We also find the method that maximizes validation AUROC while using only standard design choices. We show our results in Figure 1.

On CHEXPERT and MIMIC, we find that surprisingly even the best methods we can construct using domain-specific design choices are not significantly better than the best methods we can construct using only standard choices from image classification. Furthermore, although on CHEST X-RAY 14 the standard method is not quite as strong, its performance is still within 1.5% AUROC of the best domain-specific methods. We also note that several prior works have found performance on CheXpert and MIMIC to be more representative than performance on NIH (Pooch et al., 2020; Irvin et al., 2019). These results demonstrate the need to compare to strong baselines and also suggest that we have yet to determine how best to leverage domain-specific knowledge in chest x-ray classification.

### 3.2. Effects of Design Choices

Our results in the previous section show that domain-specific methods are likely not what drive performance in chest x-ray classification as their performance can be matched by methods using only standard image classification design choices. In an attempt to understand what design choices *do* drive performance, we consider a comparison of design choices that allows us to compute the effect of each design choice on average AUROC.

**Improving each axis of a generic ImageNet baseline.** We begin by considering a simple task: improving a generic ImageNet model. A ResNet-50 trained via BCE loss with no data augmentation is a common baseline in general image classification (Developers, 2016). Given the development of both domain-specific design choices and standard practices in chest x-ray classification, we might expect that we can improve the performance of this baseline on chest x-ray classification benchmarks substantially through changes in each axis of the model. Figure 2 shows how performance changes when we consider changing the loss function, backbone, pooling method, and data augmentation for this simple baseline model. We find there is *no change we can make for any single axis* which significantly improves performance across all three chest x-ray datasets. Moreover, one domain-specific choice—hierarchical training—consistently performs *worse* than this simple baseline.

**Comparing design choices.** In the last section, we computed the effect each design choice on a single method. However, our data allows us to also compute the average effect of a design choice across *all* methods. For each axis of a method and each design choice for that axis, we compute the difference in performance between methods using the design choice and methods using a generic choice for the axis with all other design choices fixed. We get the average effect of the design choice by averaging the paired difference over all configurations of design choices for other axes. Figure 3 shows the average effect for each design choice on each benchmark. It turns out almost no design choices have a statistically significant effect on AUROC across all three

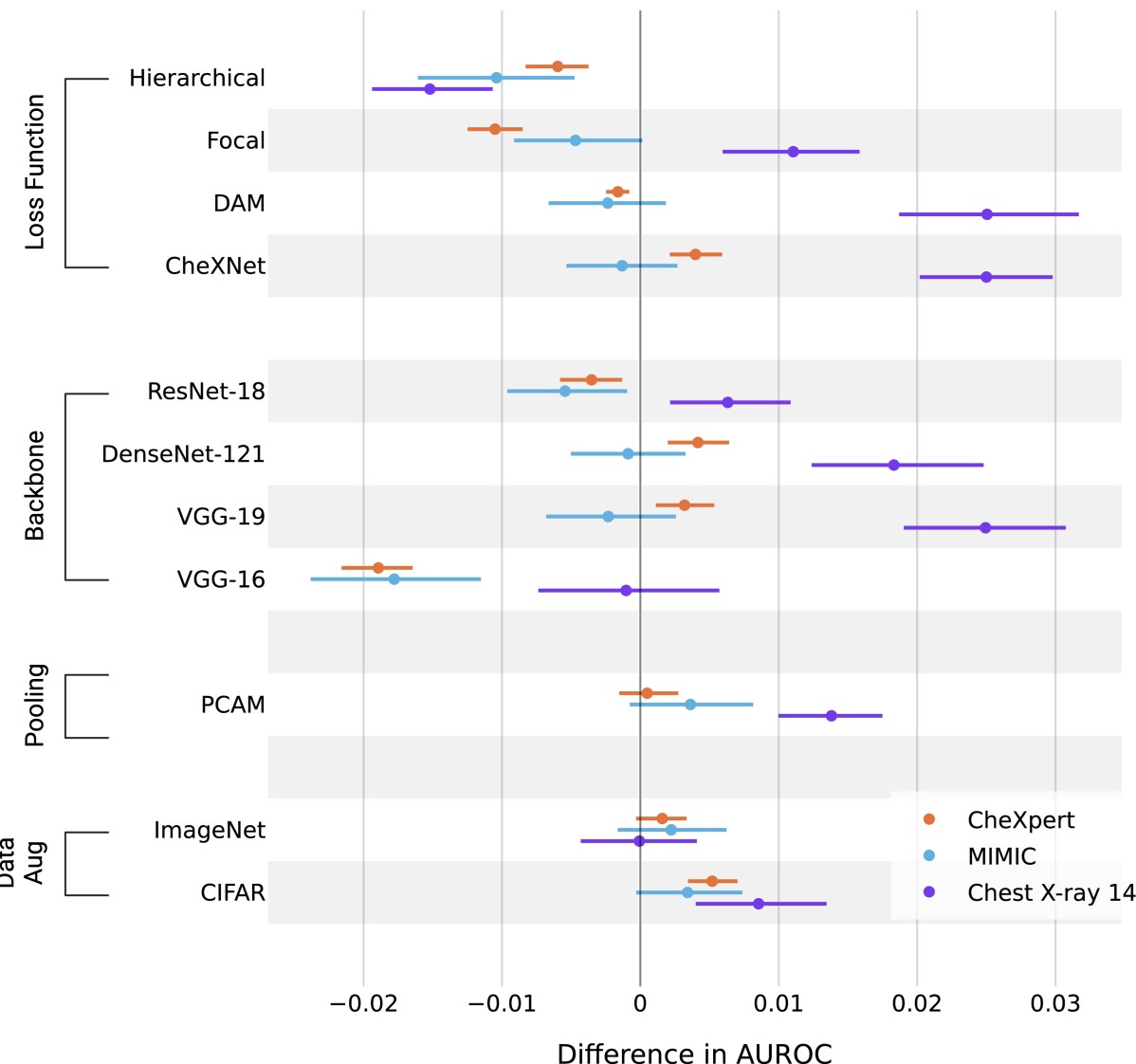

*Figure 2.* Trying to improve a generic ImageNet baseline. Each panel shows the change in performance achieved by changing *one* axis of a generic ImageNet baseline (No Augmentation, ResNet-50, BCE Loss, Standard Pooling) to a different design choice. All design choices for other axes are held fixed. We compute error bars via 1000 bootstrap samples of the test set. There is *no single change* that results in a statistically significant improvement in AUROC across all three benchmarks.

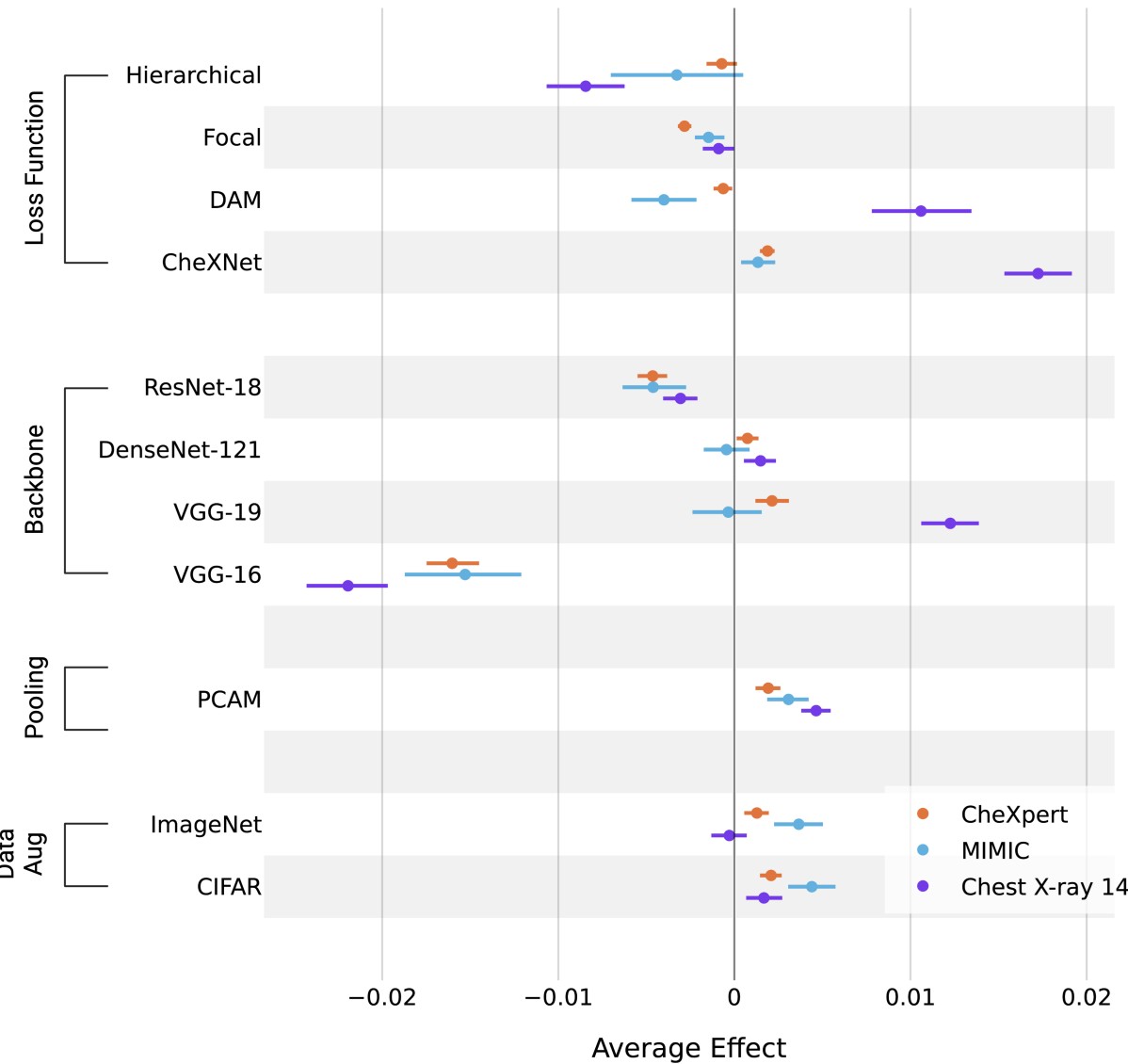

*Figure 3.* Average effect of design choices over generic choices. We conduct a controlled comparison of design choices by evaluating the average change in AUROC across all methods when we swap out a generic choice with another design choice for the same axis. Error bars are computed via the bootstrap over 1000 test sets. Generic choices are: No Augmentation, Standard Pooling, ResNet-50, and BCE Loss. Almost no design choices increase AUROC across all three datasets. Those that do (PCAM pooling, CheXNet loss type, and CIFAR data augmentation) have a very small effect (often less than 0.5% AUROC).

benchmarks. Among those choices which do, the effect is very small (often less than 0.5% AUROC). We note that the DenseNet-121 architecture—which has been observed to be a strong choice for chest x-ray classification in many past works (Irvin et al., 2019; Pham et al., 2021; Rajpurkar et al., 2017)—provides essentially no effect. These results suggest that in isolation, many design choices provide very little improvement in benchmark performance over generic choices.

### 3.3. Impact on Decision-Making

Up to this point, we have evaluated design choices using the change in average AUROC over classes. However, AUROC is a summary statistic that does not always reflect the improvement a method can offer in decision-making. To make decisions using a binary classifier which outputs a score for each sample, one has to set a threshold to mark the decision boundary between positive and negative predictions. Each threshold corresponds to a unique point on the ROC curve and can thus be specified by a false positive rate (FPR) and a true positive rate (TPR). In order to evaluate the effect of each design choice on decision-making, we show how the TPR changes at a given FPR when we swap the design choice in for a generic design choice.

For a single class, we compute the change in TPR at a given FPR by taking the average difference in TPR between methods using the design choice and methods using a generic choice in the same manner as the previous analysis. Since each dataset has multiple classes, we report the *median of the average change in TPR across classes*. This statistic allows us to report results representative of a *majority* of the classes. We compute the error bars for each point by repeating this same computation over 1000 bootstrap samples of the test set.

Figure 4 shows that for representative design choices, the improvement in TPR is almost always less than 1% for a majority of classes. These results show that even when design choices might improve AUROC in a statistically significant manner, they are unlikely to improve decision-making in a meaningful way, no matter what threshold is used. We include additional figures for other design choices in Appendix A.3.

## 4. Discussion

Our findings shed light on both how we should evaluate new methods in chest x-ray classification and which directions will help us identify strong design choices going forward.

**Best practices in chest x-ray classification.** Our analysis, shows design choices almost never consistently improve performance *across benchmarks*. That is, while they may

improve performance on one benchmark, they generally do not improve performance on other datasets. Also, even when design choices do lead to improvement, the magnitude of improvement is often so small that it is dwarfed by the statistical uncertainty in measurement. Our results therefore underline the importance of contextualizing the effect of individual improvements with both (a) uncertainty estimates and (b) performance across multiple datasets. It also seems that prior work has not always used the best hyperparameter configurations available: for example, Wang et al. (2017)—who present the first large scale chest x-ray dataset, CHEST X-RAY 14—train a baseline model using a standard architecture and loss (ResNet-50 with cross entropy loss). However, on average this model obtains 0.10 lower AUROC on CHEST X-RAY 14 than the ResNet-50 models that we train which use the same training setup. This suggests that following best practices for model selection and comparison will help us ensure that new methods represent a significant and consistent improvement over standard image-classification methods.

**Paths for improving chest x-ray models.** We have not yet identified design choices that consistently appreciably improve model performance over a generic learning setup (i.e., a ResNet-50 trained with cross-entropy loss and no data augmentation) in the chest x-ray domain. This state of affairs suggests one of two possible scenarios. The first possibility is that we have yet to discover design choices that improve chest x-ray domain performance far beyond what we could achieve with generic image classification models. The second possibility is that we *have* developed design choices that consistently improve performance, *but* existing chest x-ray benchmarks do not capture the spectrum of chest x-ray domain tasks properly—for example, the labels in these datasets might be of such a low quality that it is impossible for improvements to emerge.

## 5. Related Work

Development of domain-specific methods for chest x-ray classification has been driven by the creation of chest x-ray benchmarks. The CHEST X-RAY 14 benchmark (Wang et al., 2017) created by the NIH was one of the first chest x-ray benchmarks that allowed for the development of chest x-ray specific deep learning methods. The CHEXPERT and MIMIC benchmarks (Irvin et al., 2019; Johnson et al., 2019) improved upon the CHEST X-RAY 14 benchmark by providing more data and a better labeling process.

The design choices we evaluate are primarily derived from the best performing methods on the CHEXPERT and CHEST X-RAY 14 benchmarks. PCAM, Hierarchical training, and DAM loss all come from the top three methods on the CHEXPERT benchmark (Yuan et al., 2020; Pham et al.,

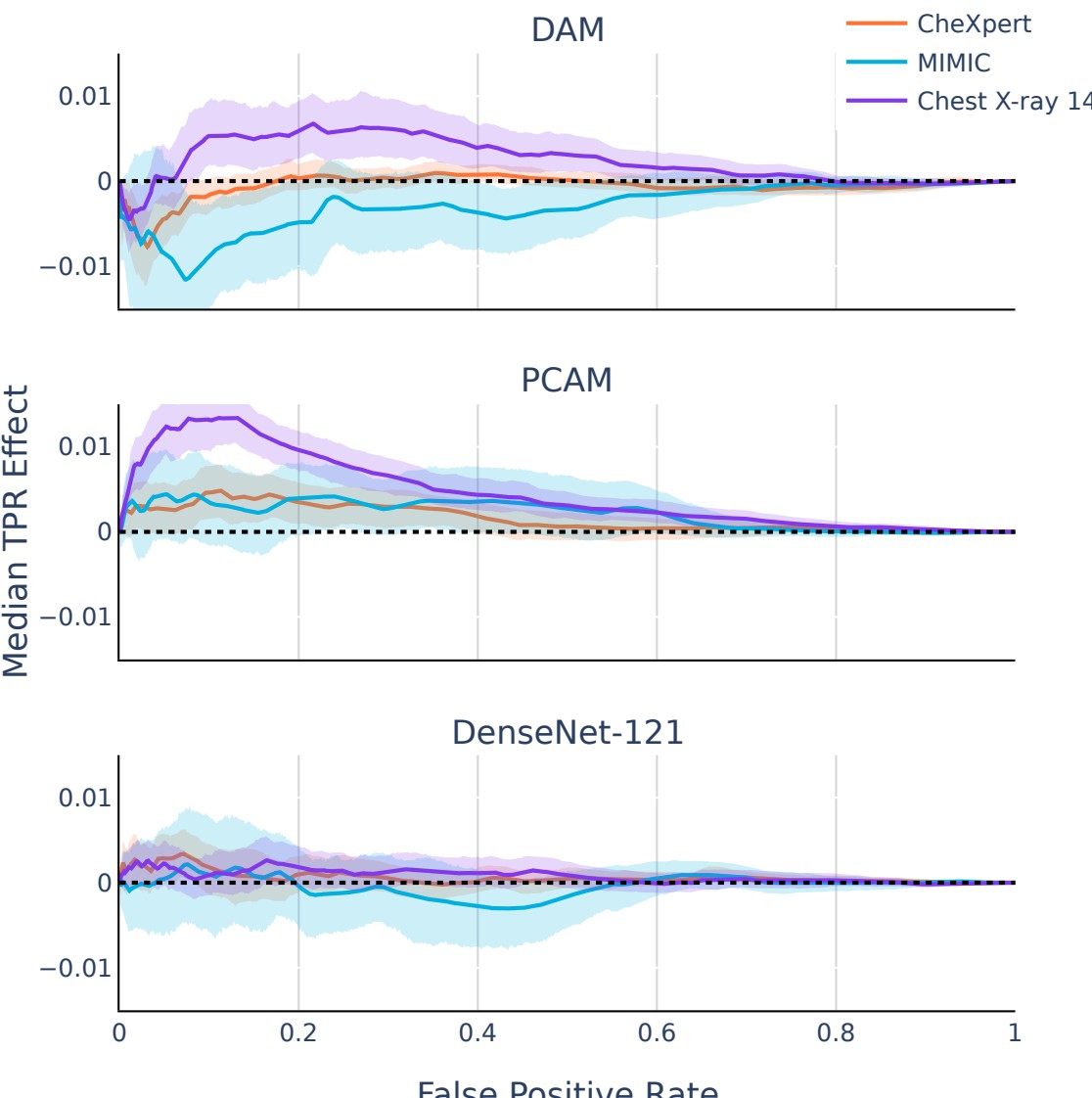

*Figure 4.* Evaluating the effect of design choices on TPR. We take three representative design choices and show their average effect on TPR at each FPR. For each class, we compute the average difference in TPR at each FPR between methods which use a given design choice and methods which use a generic choice for the same axis, keeping all choices fixed along other axes. We then take the median across classes of this average difference. The process is repeated 1000 times over bootstrap test sets to derive confidence intervals. We see that the effect of these design choices on TPR is almost always below 1% for a majority of classes. We include additional figures for other design choices in Appendix A.3

2021; Ye et al., 2020), and CheXNet comes from a top performing method on the CHEST X-RAY 14 benchmark (Rajpurkar et al., 2017). Other design choices we evaluated are standard choices in image classification that have also seen success in the chest x-ray domain (Moses, 2021).

Prior works have evaluated models across multiple chest x-ray datasets and found disparate performance (Pooch et al., 2020). Some have also observed that simple models do best when evaluating other aspects of chest x-ray classification models such as their fairness (Zhang et al., 2022). Ke et al. (2021) did an evaluation of one axis we consider–backbones–but only on the CheXpert dataset. To the best of our knowledge no prior work has directly evaluated the effect of a wide range of design choices on chest x-ray classification performance across multiple benchmarks.

## 6. Conclusion

By evaluating the effect of a wide range of design choices across several canonical chest x-ray benchmarks, we aim to shed light on what design choices drive performance in chest x-ray classification. We find that methods designed specifically for the chest x-ray domain (*domain-specific* methods) are often matched by standard methods in computer vision. More broadly, very few design choices consistently exhibit significantly better performance than those of a generic image classification model. Our results demonstrate the importance of comparing new methods to strong baselines across multiple benchmarks with robust uncertainty evaluation. Moreover, our analysis suggests that either the methods or the benchmarks of chest x-ray classification require additional development before we can confidently identify the best design choices for this domain.

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

# A. Appendix

## A.1. Hierarchies for CHEXPERT, MIMIC, and CHEST X-RAY 14

Our hierarchy for CHEXPERT and MIMIC is taken from Pham et al. (2021) who in turn use the hierarchy given in Irvin et al. (2019). For CHEST X-RAY 14 we use our own hierarchy derived from our knowledge of the dataset. We describe each hierarchy below by giving the parent-child relationships. If $A \rightarrow B$ it means that $A$ is a parent of $B$ in the hierarchy. Classes with no parents are assumed to have their parent as the root node.

### A.1.1. CHEXPERT AND MIMIC

Enlarged Cardiomediastinum → Cardiomegaly

Lung Opacity → Edema

Lung Opacity → Consolidation

Lung Opacity → Pneumonia

Consolidation → Pneumonia

Lung Opacity → Atelectasis

Lung Opacity → Lesion

### A.1.2. CHEST X-RAY 14

Pneumonia → Atelectasis

Pneumonia → Effusion

Nodule → Mass

Pneumonia → Consolidation

Pneumonia → Edema

Fibrosis → Pleural Thickening

## A.2. Training Details

We implement all methods in PyTorch (Developers, 2016). Except where otherwise noted, we use SGD with a linear decay from the initial learning rate to 0 over the duration of training and grid search over a training duration of 1, 3, or 5 epochs. We also grid search over learning rate and weight decay using a 6x6 grid ($10^{-4}$ to $10^{-2}$ for learning rate and $10^{-6}$ to $10^{-4}$ for weight decay). We use a batch size of 256 for all methods.

Below we describe any details specific to each of the design choices we consider.

1. **Loss Function**

   (a) BCE: No special considerations.
   (b) Focal Loss: We also grid search over $\gamma$ in $[0.5, 2.0, 5.0]$.
   (c) CheXNet: We compute class weights from the train set and use the same setup as described above.
   (d) Hierarchical Loss: We use the training setup described above for the pre-training step. Taking the best hyperparameters for each method (computed via validation AUROC) we then fine tune the pre-trained model using the loss function of (Chen et al., 2019) and replace the final layer before fine-tuning. We do *not* freeze the pre-trained backbone as we found results were significantly worse when we did. We use the same grid search described above for the fine-tuning step.
   (e) DAM Loss: Following the authors' example [2], we first take the best BCE model for each combination of Data Augmentation, Pooling, and Backbone as a pretrained model. We then create one fine-tuned model per class

---

[2] https://github.com/Optimization-AI/LibAUC/blob/main/examples/05_Optimizing_AUROC_Loss_with_DenseNet121_on_CheXpert.ipynb

using their open-source loss function with the DAM-specific hyperparmeters given in their example. Due to computational burden we fine-tune all models for 2 epochs rather than grid searching over training duration. We use a constant learning rate for all epochs. Prediction is done using the fine-tuned model for each class.

2. **Backbone**: No special considerations.

3. **Pooling**: No special considerations.

4. **Data Augmentation**

   (a) No Augmentation: No special considerations.
   (b) CIFAR: We randomly translate by 10% of the image size at maximum. We use the cutout implementation of the MosaicML Composer library.
   (c) ImageNet: We randomly crop with a scale between $0.5$ and $1.0$. We use color jitter with a brightness factor of $0.2$, a contrast factor of $0.2$, a saturation factor of $0.2$, and a hue factor of $0.3$. We use the mixup implementation of the MosaicML Composer library.

We train all methods on NVIDIA A100 GPUs. We train models on a cluster of 80 gpus over a period of about 3 months. To reduce computational load we prune all runs that cannot achieve an average of 0.6 AUROC on the validation set after 2 epochs. These intensive grid searches ensure that for every method we consider we get close to the best possible performance that the method can achieve on each dataset.

The CheXpert dataset uses different labeling processes for its training and validation splits. To avoid distributional mismatch between the data we train on and the data we use for hyperparameter tuning we randomly re-split the CheXpert train set, keeping 80% of the original for training and using 10% as a validation set and 10% as a test set. We ensure that splitting is done so as to avoid overlap between patients across the partitions. For the MIMIC-CXR and Chest X-Ray 14 datasets, we use the given validation and test sets as they do not have this distributional mismatch issue. For both CheXpert and MIMIC-CXR we ignore uncertainty labels (the U-IGNORE strategy in Irvin et al. (2019)) for ease of implementation and comparison with the Chest X-ray 14 dataset which does not have uncertainty labels. All data is resized to 224 x 224 pixels. We normalize all images using the mean and standard deviation of the training dataset. For ease of implementation we copy each image to have 3 repeated channels of the same image.

On all datasets we evaluate performance using AUROC as is standard in chest x-ray classification. On CheXpert and MIMIC-CXR, we use the average performance of the 5 classes used in the test set for the CheXpert competition as our score for a method. On Chest X-ray 14, we use the average AUROC across all 14 labels. To evaluate uncertainty in our scores we recompute the test AUROC of each method using 1000 bootstrap samples over the test partition for each dataset.

### A.3. Plots of Effect on TPR at a given FPR

Each plot shows the *median* effect of a design choice on TPR at a given FPR where the median is computed over classes. Confidence intervals are given by the 2.5th and 97.5th percentiles of the distribution of the median effect computed over 1000 bootstrap samples. The effect of TPR at a given FPR for a single class is given by the average difference in TPR across all methods that have the design choice and all methods that have a generic choice for the same axis. For example if the design choice is DAM Loss we would compute—for every combination of data augmentation, pooling, and backbone—the difference in TPR between the method when it uses DAM loss and the method when it uses BCE Loss. We would then average this difference over all combinations of data augmentation, pooling, and backbone. The caption for each figure gives the design choice in question. Loss functions are compared to the generic BCE loss. Backbones are compared to the generic ResNet-50 backbone. PCAM is compared to standard pooling, and data augmentations are compared to the generic No Augmentation.

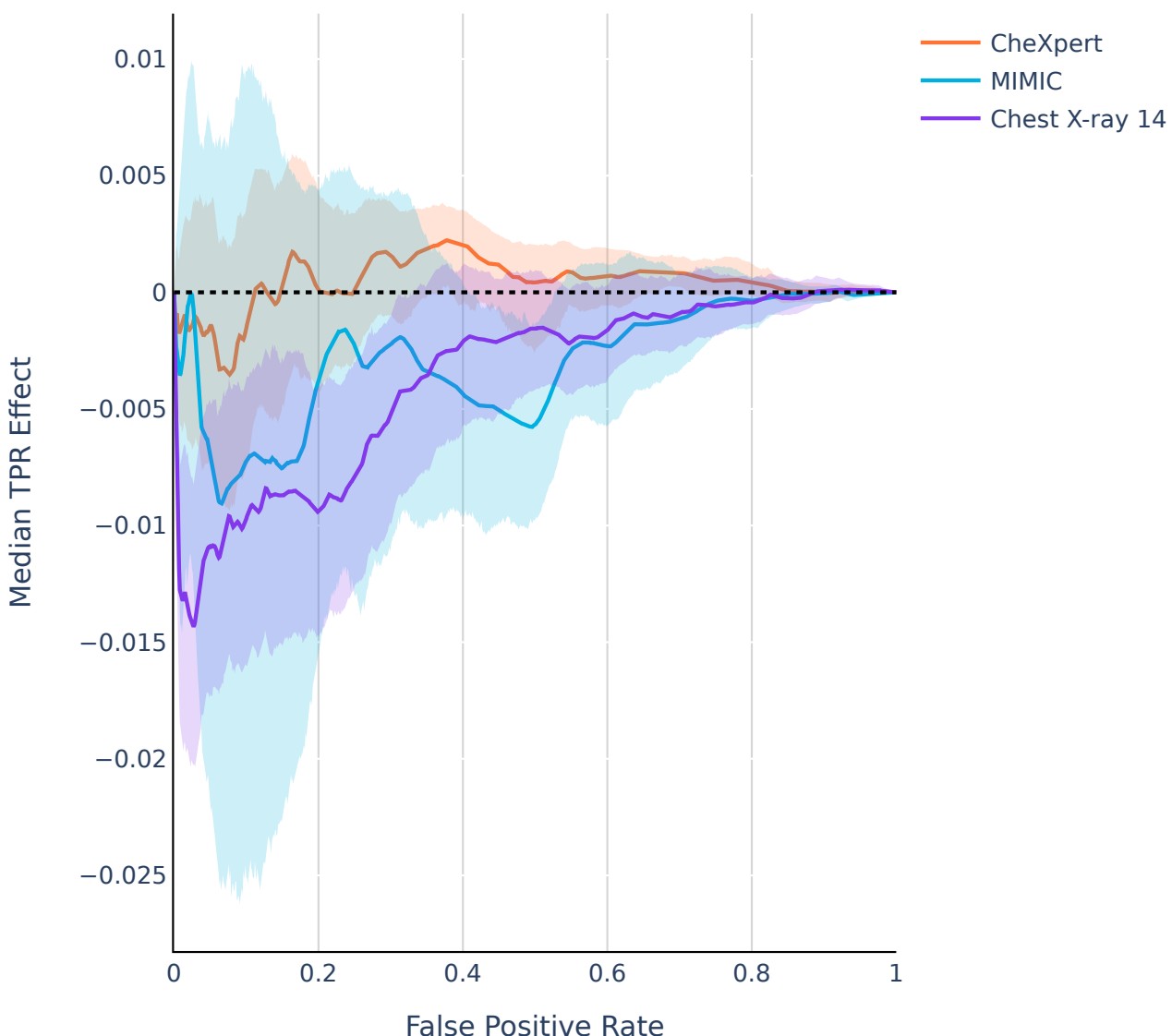

*Figure 5.* Loss Function: Hierarchical Loss Function

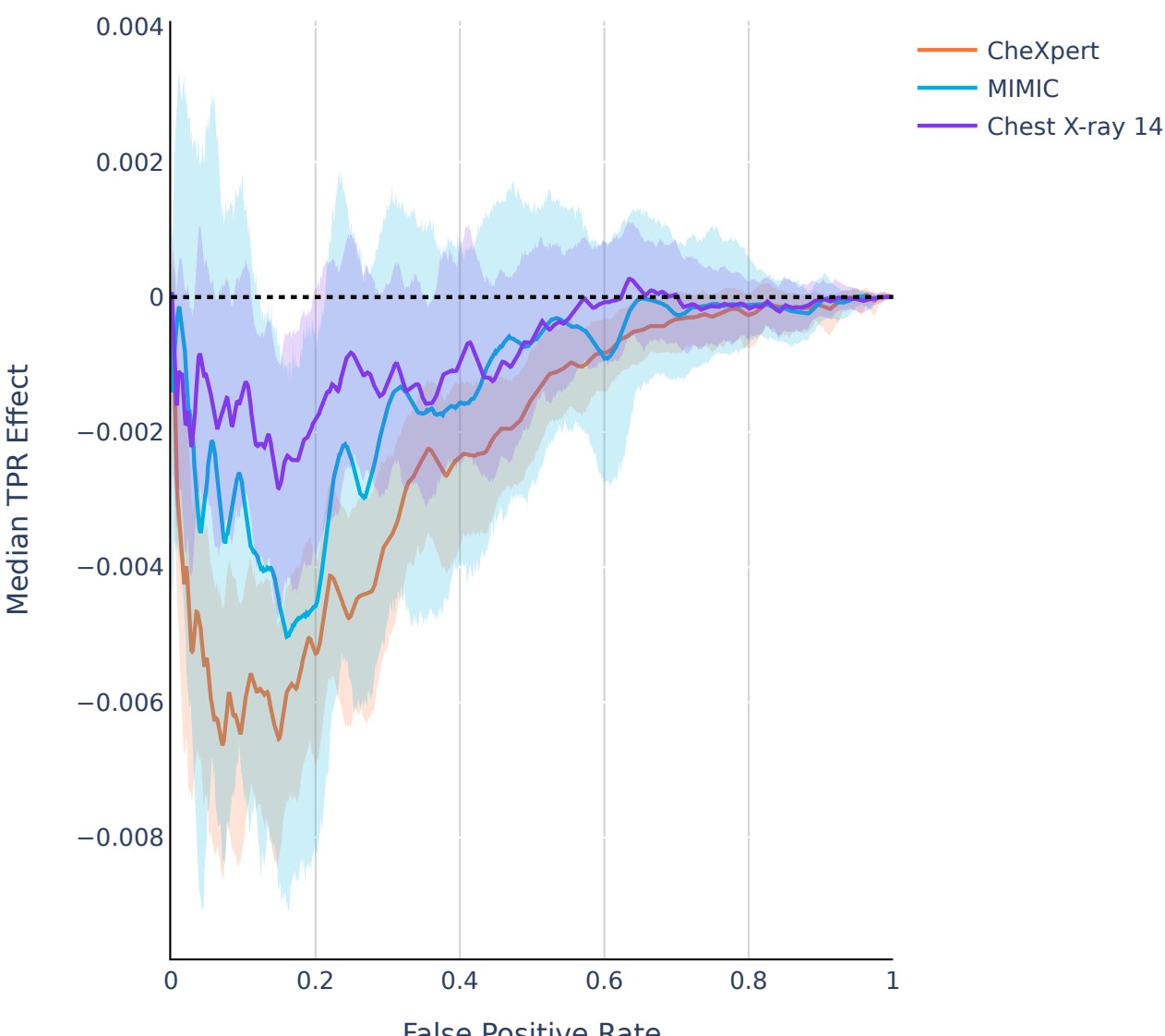

*Figure 6.* Loss Function: Focal Loss Function

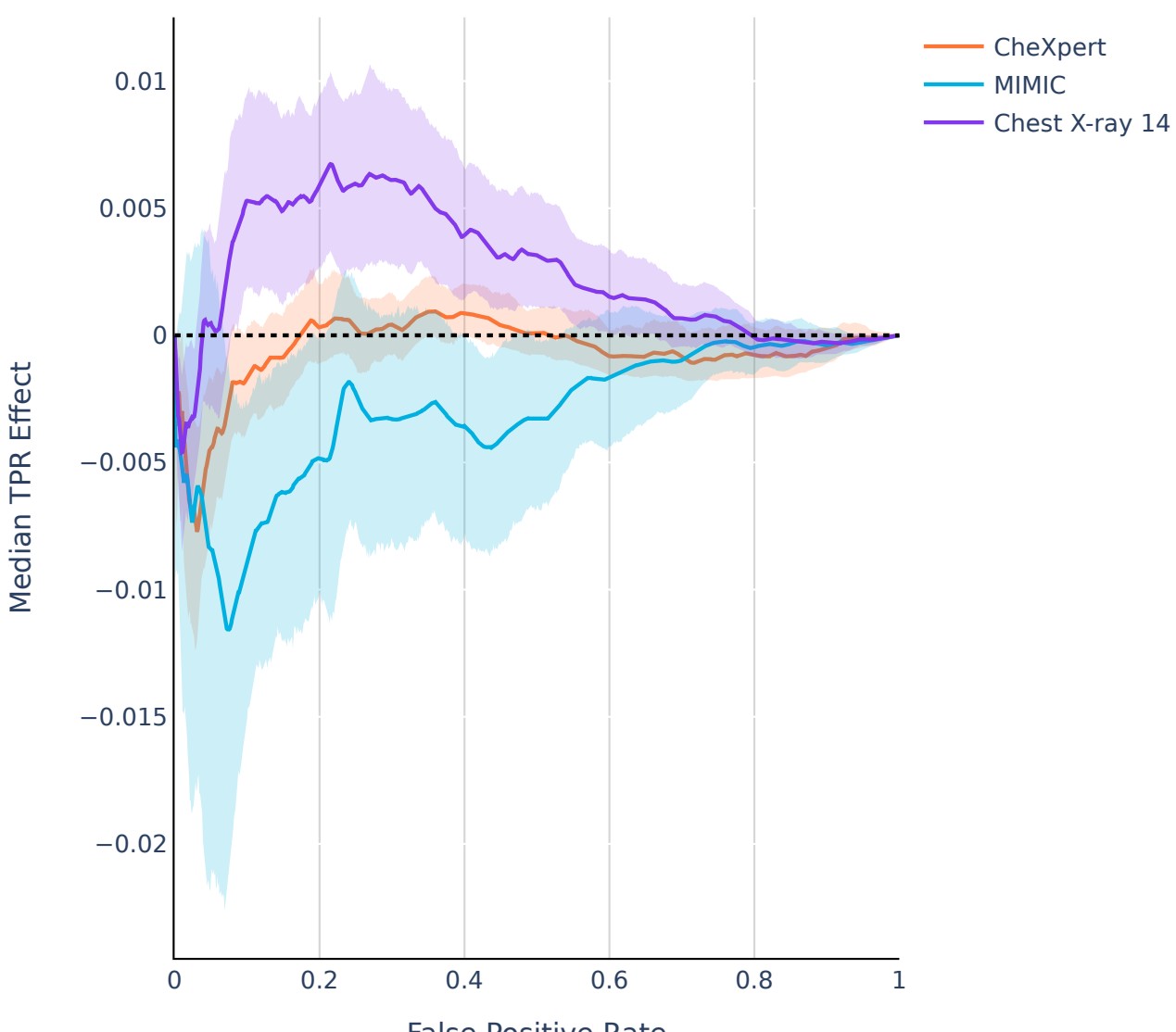

*Figure 7.* Loss Function: DAM Loss Function

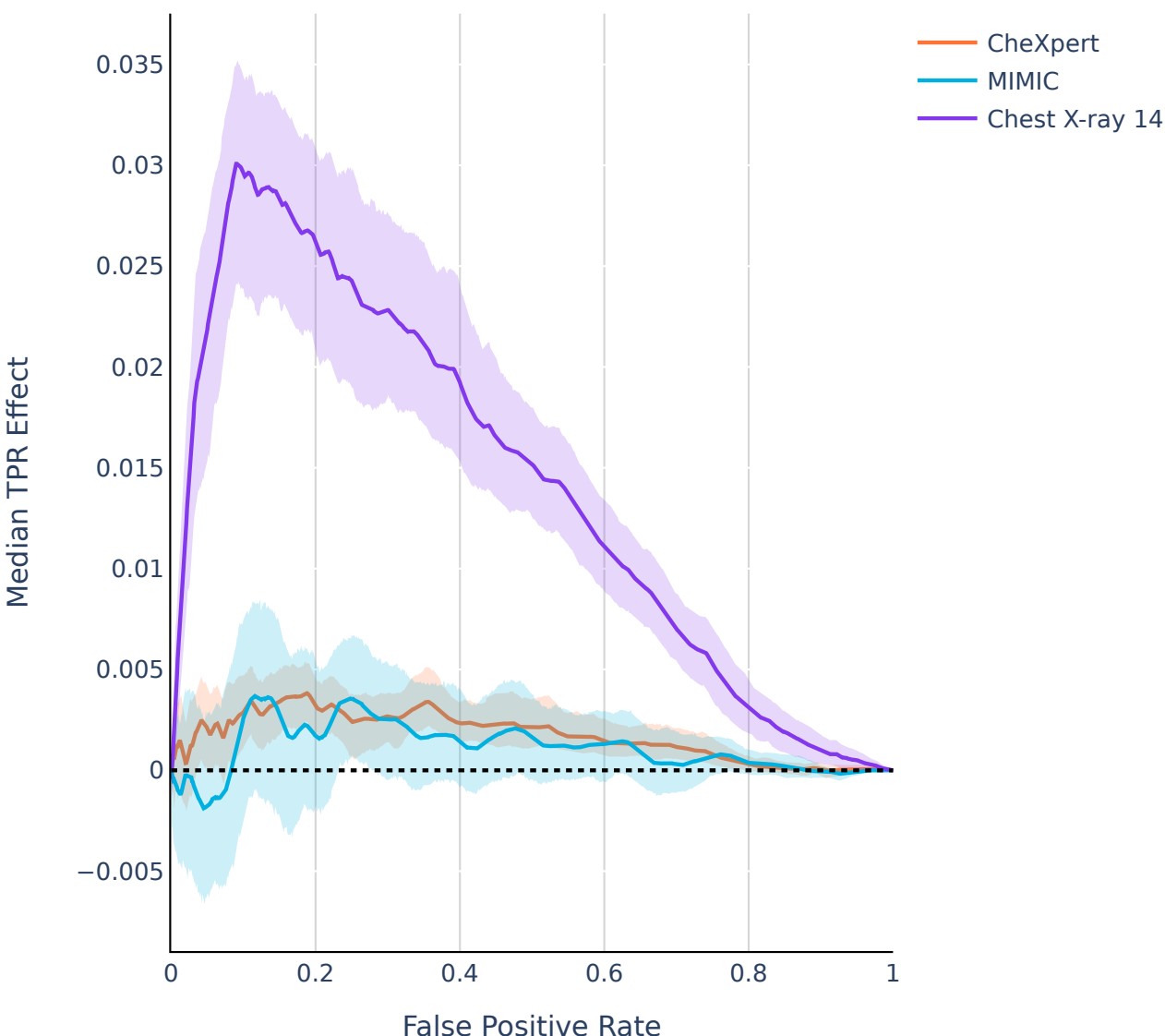

*Figure 8.* Loss Function: CheXNext Loss Function

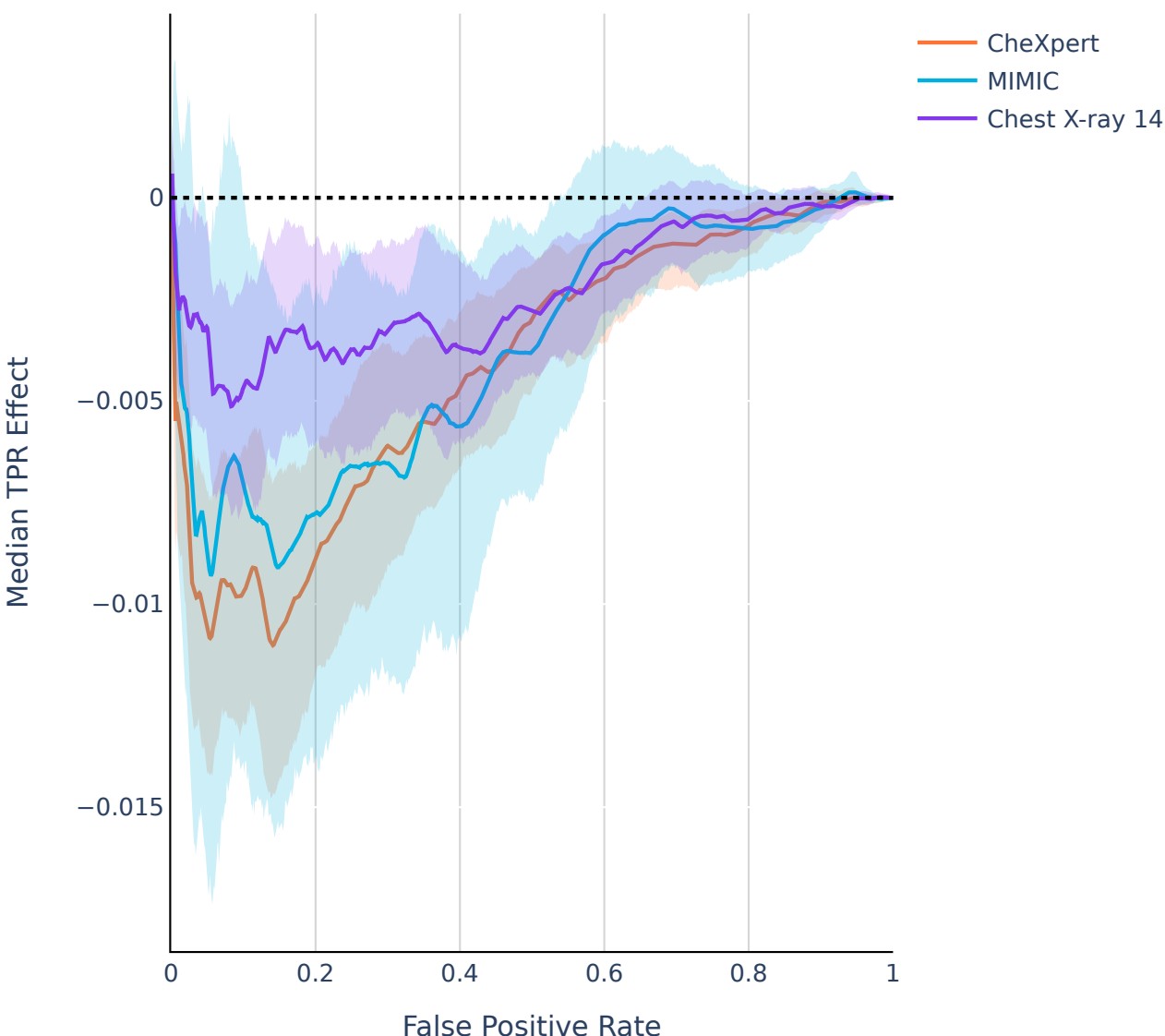

*Figure 9.* Backbone: ResNet-18

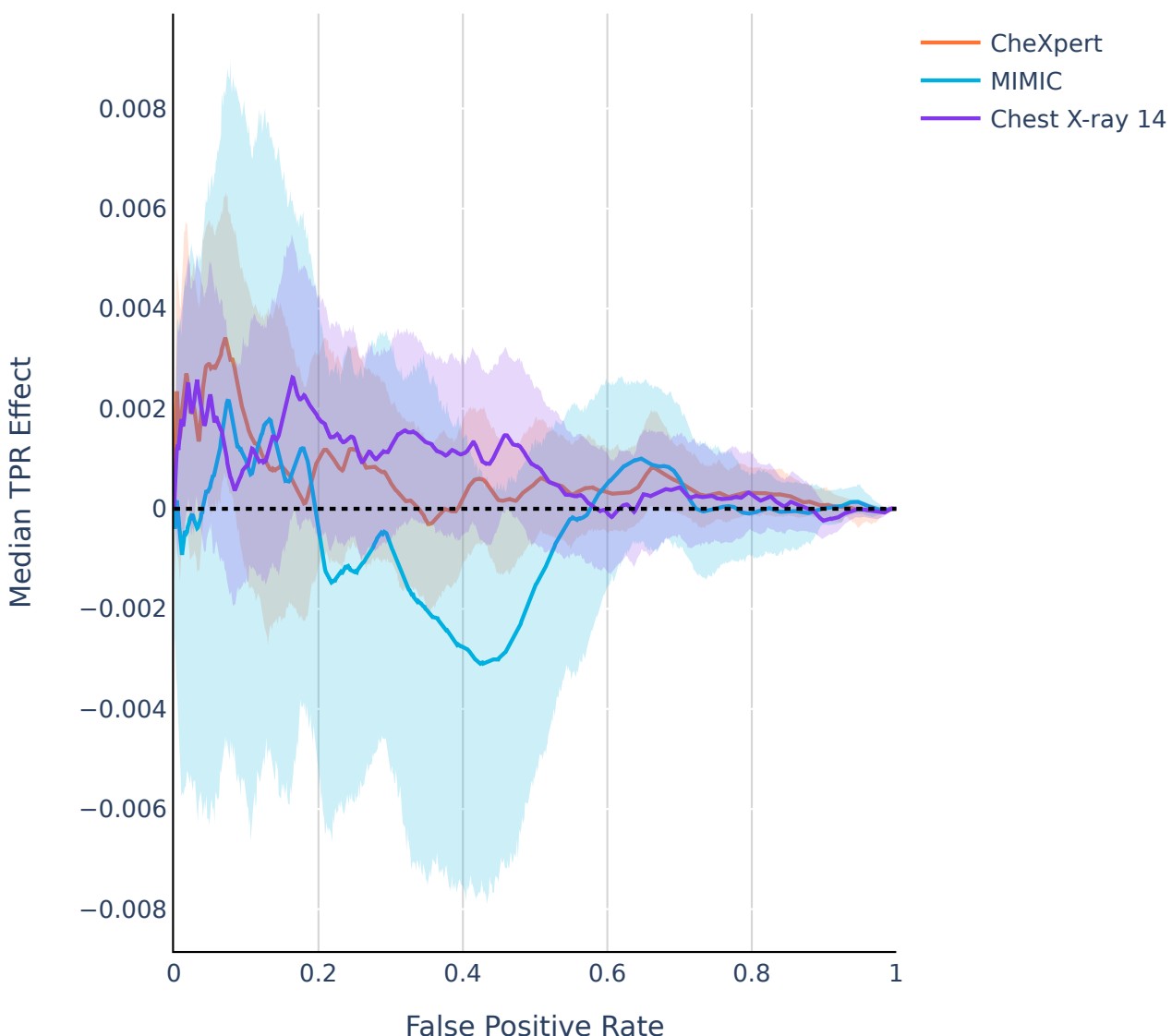

*Figure 10.* Backbone: DenseNet-121

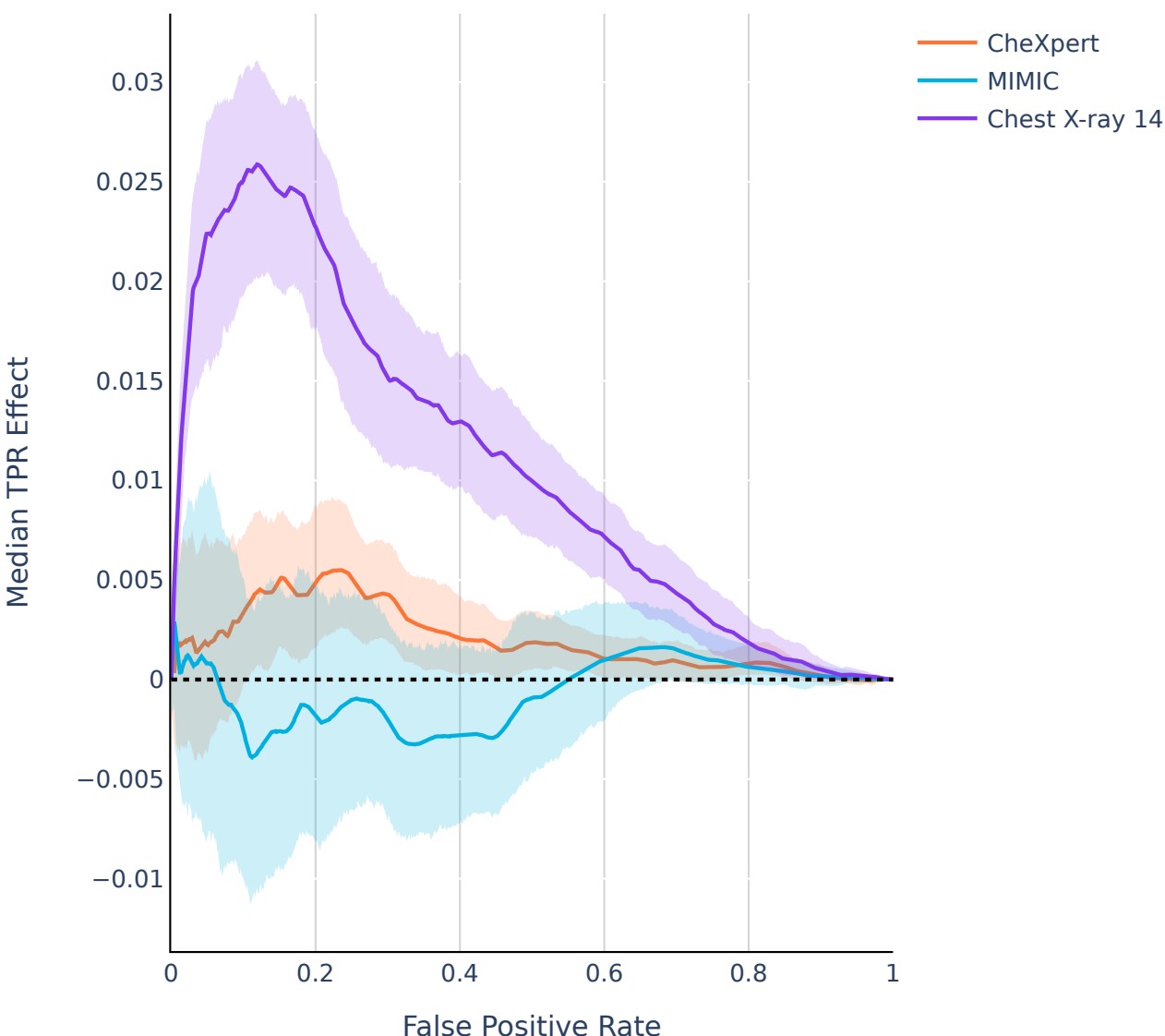

*Figure 11.* Backbone: VGG-19 With Batch Norm

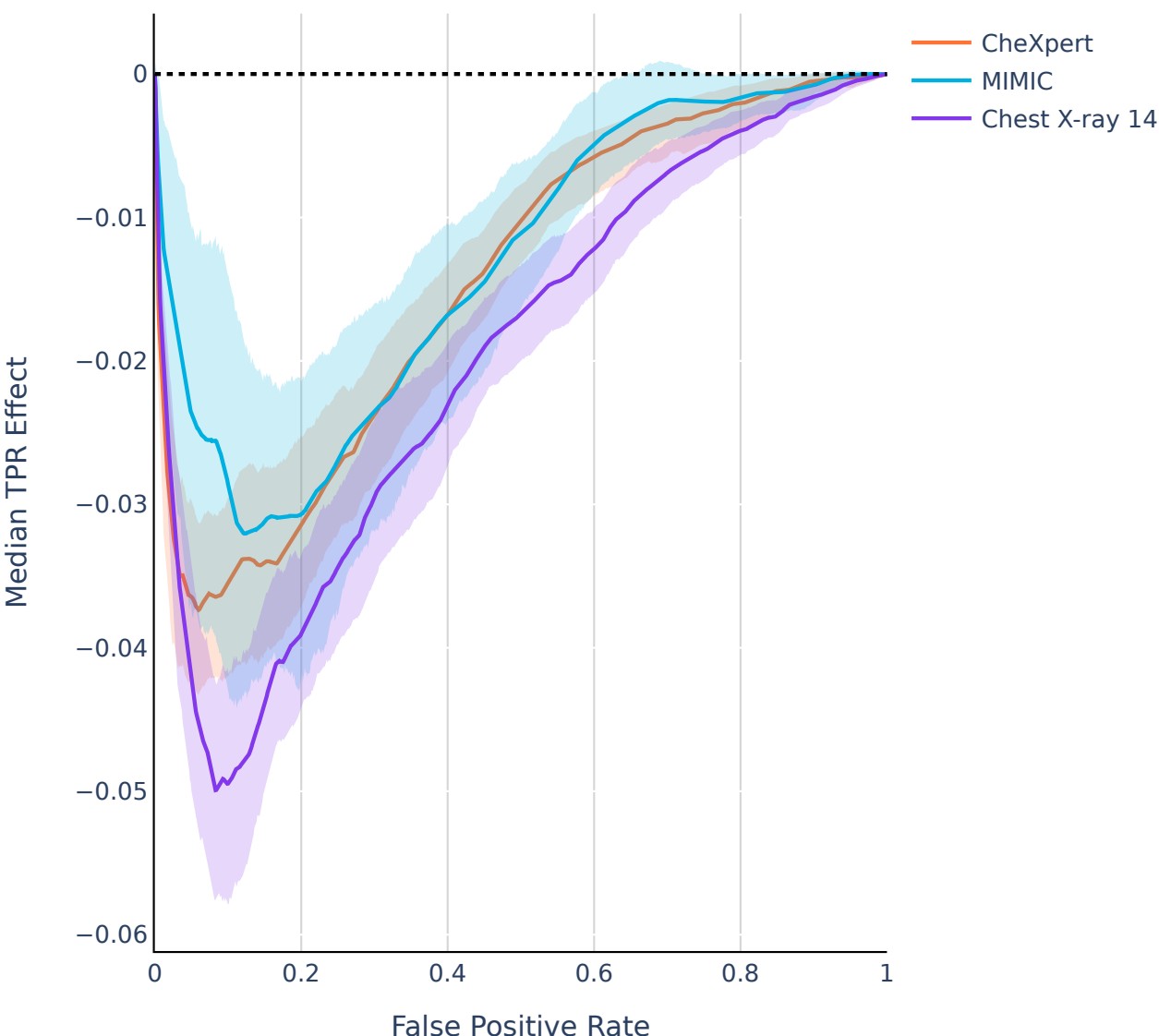

*Figure 12.* Backbone: VGG-16

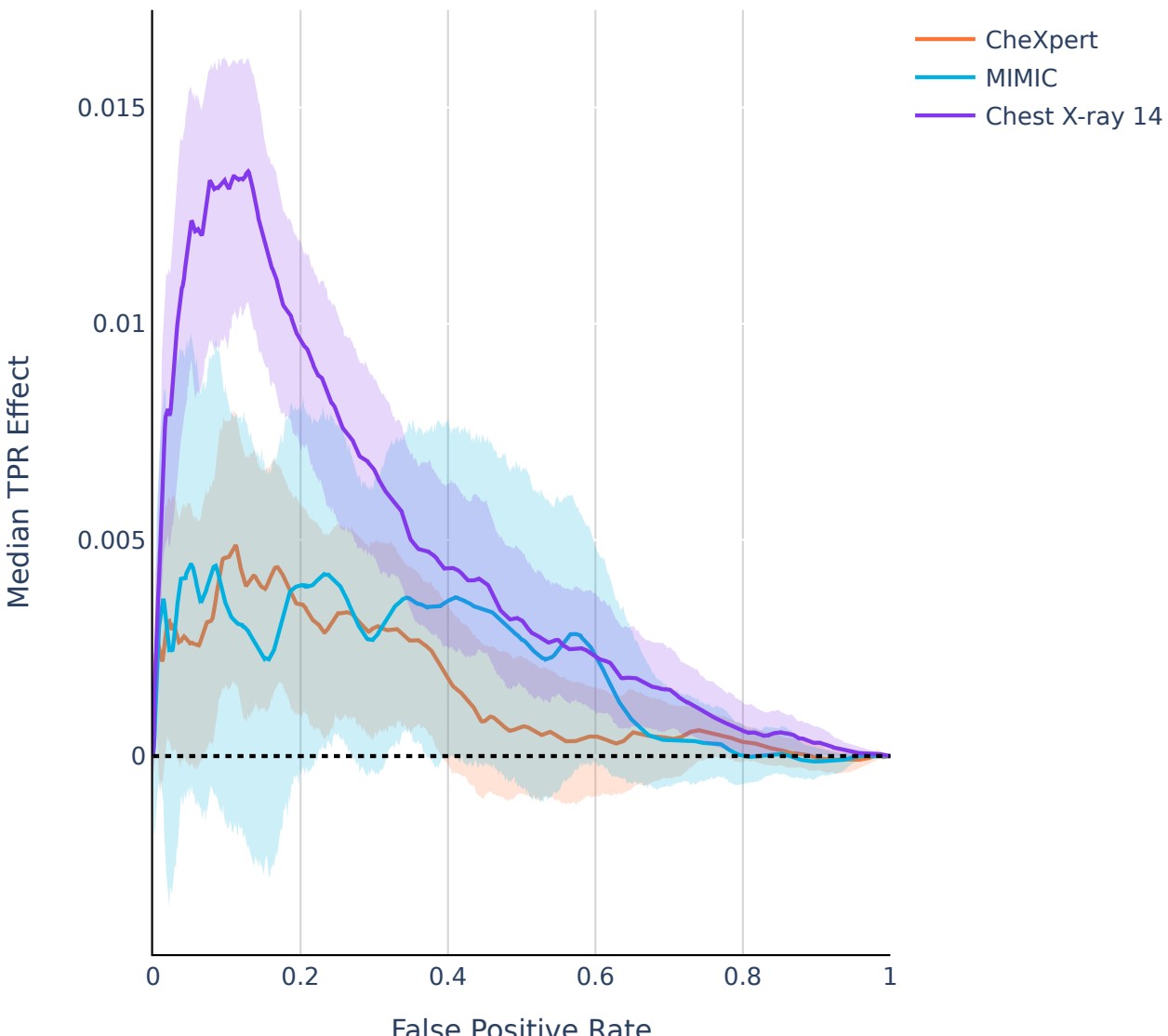

*Figure 13.* Pooling: PCAM

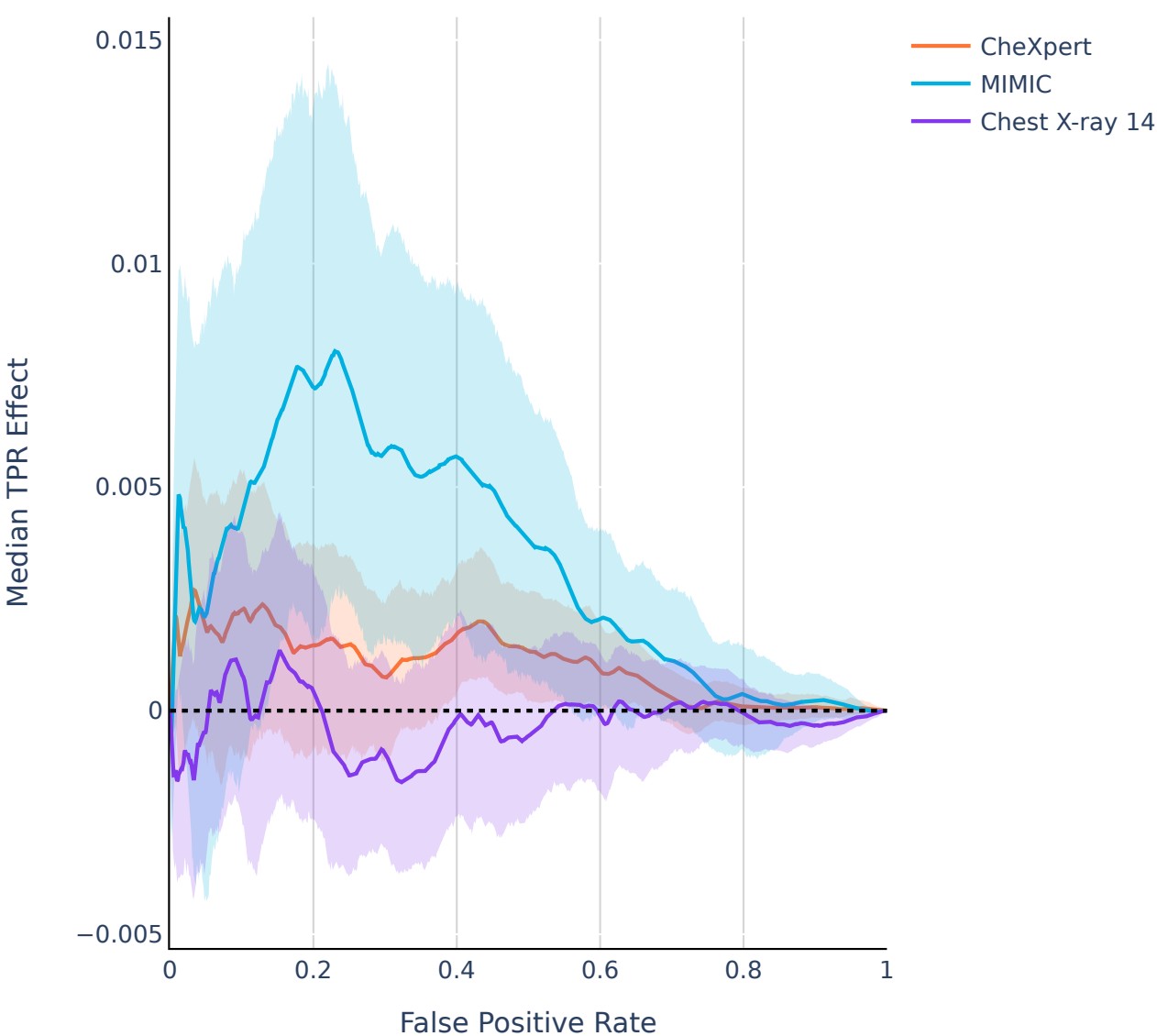

*Figure 14.* Data Augmentation: ImageNet Data Aug.

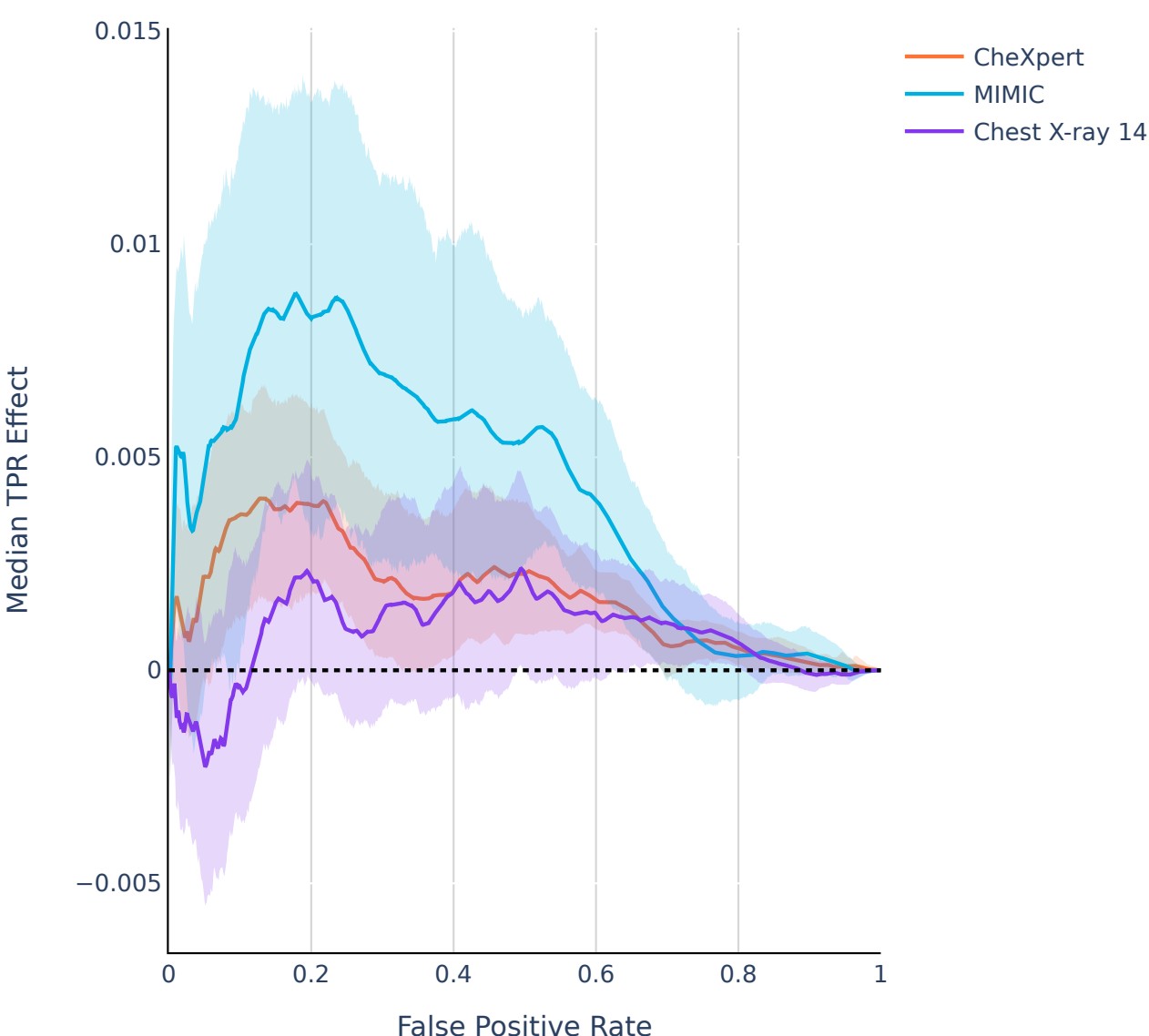

*Figure 15.* Data Augmentation: CIFAR-10 Data Aug.

