# OpenReview forum: "What Works in Chest X-Ray Classification? A Case Study of Design Choices"
_ICML.cc/2023/Workshop/IMLH — IMLH 2023 Poster_

### Official Review · Reviewer_cBLV · 2023-06-16
**Investigate the effect of several design choices for chest x-ray**

**Rating:** 5
**Confidence:** 4

**Review:**

Summary and Strength:
The paper addresses the issue of non-standardized training setups in chest x-ray prediction models, which use specialized design choices for the chest x-ray domain, making it unclear how these individual design choices affect performance. The authors examine a wide range of model design choices on three canonical chest x-ray benchmarks and find that a properly tuned model composed of standard image classification design choices can match the performance of even the best domain-specific models. The authors also discover that none of the proposed design choices, including commonly used choices like the DenseNet-121 architecture or basic data augmentation, consistently improve performance over a generic learning setup consisting of a barebones ResNet-50 with cross-entropy loss and no data augmentation. Overall, the results suggest that leveraging a properly tuned model composed of standard image classification design choices can achieve comparable performance to domain-specific models in chest x-ray prediction tasks.

Weakness:
- The paper may have limited novelty as it only investigate several factors for X-ray prediction task. However, a more comprehensive investigation has been done in paper 'Why is the winner the best?' [1].

[1] Eisenmann, Matthias, et al. "Why is the winner the best?." Proceedings of the IEEE/CVF Conference on Computer Vision and Pattern Recognition. 2023.

---

### Official Review · Reviewer_yF1A · 2023-06-18
**The finding is interesting.**

**Rating:** 6
**Confidence:** 5

**Review:**

The author compared different methods and found that current classification models specifically designed for chest X-rays do not necessarily outperform general-purpose architectures, which is quite interesting.
It would be more clear for the legends (Fig.2 and Fig.3) in the paper to indicate which models are specific to the domain and which ones are general-purpose when comparing different methods.

---

### Meta-Review · Area_Chair_XLjL · 2023-06-20

**Recommendation:** Accept (Poster)
**Confidence:** 5

**Metareview:**

The authors discussed empirically designed studies on Chest X-Ray Classification. This paper designed case studies on different choices of models, training setups, augmentation, loss functions, etc. The individualized studies make the paper contribute to the Chest X-Ray classification problems. The community can potentially benefit from these empirical findings.

---

### Decision · Program_Chairs · 2023-06-20

Accept (Poster)